# Superplastic nanoscale pore shaping by ion irradiation

Morteza Aramesh [1,2,3], Yashar Mayamei[4], Annalena Wolff[5] & Kostya (Ken) Ostrikov[1,2]

Exposed to ionizing radiation, nanomaterials often undergo unusual transformations compared to their bulk form. However, atomic-level mechanisms of such transformations are largely unknown. This work visualizes and quantifies nanopore shrinkage in nanoporous alumina subjected to low-energy ion beams in a helium ion microscope. Mass transport in porous alumina is thus simultaneously induced and imaged with nanoscale precision, thereby relating nanoscale interactions to mesoscopic deformations. The interplay between chemical bonds, disorders, and ionization-induced transformations is analyzed. It is found that irradiation-induced diffusion is responsible for mass transport and that the ionization affects mobility of diffusive entities. The extraordinary room temperature superplasticity of the normally brittle alumina is discovered. These findings enable the effective manipulation of chemical bonds and structural order by nanoscale ion-matter interactions to produce mesoscopic structures with nanometer precision, such as ultra-high density arrays of sub-10-nm pores with or without the accompanying controlled plastic deformations.

[1] School of Chemistry, Physics and Mechanical Engineering and Institute for Future Environments, Queensland University of Technology (QUT), Brisbane QLD 4000, Australia. [2] CSIRO-QUT Joint Sustainable Processes and Devices Laboratory, Common wealth Scientific and Industrial Research Organisation, Lindfield NSW 2070, Australia. [3] Laboratory of Biosensors and Bioelectronics, Institute for Biomedical Engineering, ETH Zürich, 8092 Zürich, Switzerland. [4] Department of Nano Science, University of Science and Technology, Daejeon 34113, Republic of Korea. [5] Central Analytical Research Facility, Institute for Future Environments, Queensland University of Technology (QUT), Brisbane QLD 4000, Australia. Correspondence and requests for materials should be addressed to M.A. (email: mrtz.aramesh@gmail.com)

Low-energy ion beams can be implemented to fashion matter at nano-dimensions[1]. The "nano" size of matter imposes ion-matter interactions that have not been visualized in bulk materials[2,3]. The impact of low-energy ions induces redistribution of target molecules and subsequently re-shaping of the material in an extremely fine manner, possibly with nanometer precision[4]. Modification and shaping of nanomaterials using low-energy ion beams is thus important from both fundamental and technological perspectives.

The interaction of low-energy light ions with materials are often neglected because energetic interactions through primary knock-on atoms are less dominant (compared to heavier ions such as Ga$^+$) and the ionization effects are less profound compared to swift heavy-ion beams. However, several experiments have recently demonstrated that these interactions are very significant at nanoscales[4–6]. For instance, the size of a prefabricated pore (in a silicon nitride membrane) can be reduced to a sub-nanometer domain when irradiated with keV Ar$^+$ ions[1], or the length of prefabricated nanowires (GaAs and InAs) can be increased by keV He$^+$ irradiation[4]. Various aspects of interaction of low-energy light ions with nanomaterials are being investigated[7–9]. However, the role of the chemical bonds and degree of disorder in bond breaking and diffusion of matter subjected to irradiation remains to be elucidated[10–12]. The outcomes of interplay between bond breaking and diffusion at atomic scales largely determines material reconstruction at microscales. An innovative approach toward better understanding of this interplay is to visualize mass transport at nanoscales[4].

In this study, helium ion microscope (HIM – ORION NanoFab) is used to simultaneously induce, visualize, and quantify the dynamics of nanoscale transformation in nanoporous alumina arrays subjected to low-energy ion beams. We aim to answer the question 'how could nanoscale and microscale response of a material to ion irradiation be manipulated at atomic level?'. The results provide a perspective to the phenomena by correlating the atomic structure to nanoscopic response and the subsequent microscopic behavior. These findings reveal potentially deterministic strategies for nanomatter shaping by manipulating nanoscale ion-matter interactions. Thin films of nanoporous anodic aluminum oxide (AAO), obtained by the electric field-assisted oxidation of Aluminum films[13] is chosen deliberately as the proof-of-principle platform. Re-shaping of the nanopores is used to visualize and quantify mass transport. This process is resolved with atomic resolution, through the interface analysis using transmission electron microscopy (TEM). Also, due to its tuneable chemistry (ionic vs covalent bonds), purity, and crystallinity[13], AAO enables the study of the role of chemical bonds and disorders in ion-matter interactions. The as-prepared AAO membranes have amorphous atomic structure (am-AAO), while the crystalline AAO membranes (c-AAO) are produced by annealing the amorphous films at higher temperatures (up to 800 °C). The dielectric nature of the AAO films makes it possible to induce the effects of ionization and electric charge accumulation on the nanoscale dimensions of the pore arrays. In this way, this study thus connects the observed mesoscale material deformations with atomic-level reconstructions through nanoscale pore re-shaping under various irradiation and ionization conditions. We also discover the extraordinary superplasticity of the normally brittle porous alumina at room temperature and under specific ion irradiation and bond/disorder conditions.

## Results

**System of study.** The experimental setup is depicted in Fig. 1a, where the impinging helium ions on nanoporous AAO thin membranes (either free-standing or on top of a substrate) generate secondary electrons, which are collected by a detector allowing instantaneous imaging with nanometer precision (also see Supplementary Methods). It was observed that a number of parameters influence the ion-matter interactions and mass transport in He$^+$ irradiated AAO (depicted symbolically in Fig. 1b,c). Amongst the most influential parameters were (i) the ion-beam flux ("low-flux" vs "high-flux") and (ii) chemical bonding in AAO (am-AAO vs c-AAO). In amorphous AAO (am-AAO), two significantly different flux-dependent regimes were observed during the exposure to the Helium ions: (i) At lower flux regime, the pores of am-AAO started to shrink and (ii) pore shrinkage was not observed at high-flux regime, rather am-AAO showed surface expansion due to substrate swelling. On the other hand, c-AAO was less susceptible to variations in irradiation flux; neither pore closure nor high elasticity was observed for c-AAO in low-flux and high-flux regimes, respectively. Fig. 2 summarizes the key experimental observations in amorphous and crystalline AAO irradiated with the flux of F < 10 and F > 100 ions nm$^{-2}$ s$^{-1}$, referred herein as "low-flux" and "high-flux" regimes, respectively (the term "flux" here is defined as the rate of the impinging ions per scan unit).

**Pore Closure.** Supplementary Movie 1 shows an array of nanopores in am-AAO during irradiation with low-flux 25 keV He$^+$. Ultra-high-density (~$10^{11}$ pores cm$^{-2}$) ordered arrays of sub-10

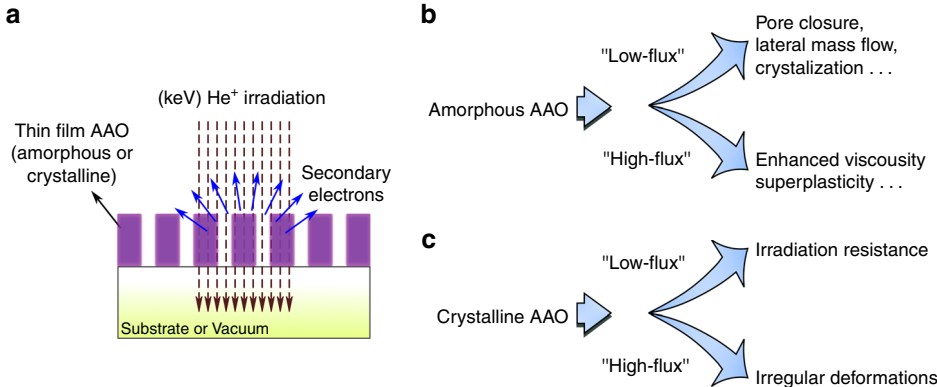

**Fig. 1** Interaction of helium ions with nanoporous AAO. **a** Shows the principles of the experimental setup. Helium ions with keV energy are generated in a gas field ion source and then are scanned over the sample in a high vacuum chamber. Secondary electrons that are generated by the sample are used for in situ monitoring of the ion-beam interactions with <1 nm resolution. The studied samples are thin films of nanoporous membranes of aluminum oxide (either supported by a substrate or locally stand free in vacuum). **b, c** Symbolic illustration of observed phenomena: the microscopic response of the matter to irradiation is determined by the interplay between the nature of the chemical bonds, ionization effects, and irradiation-induced diffusion

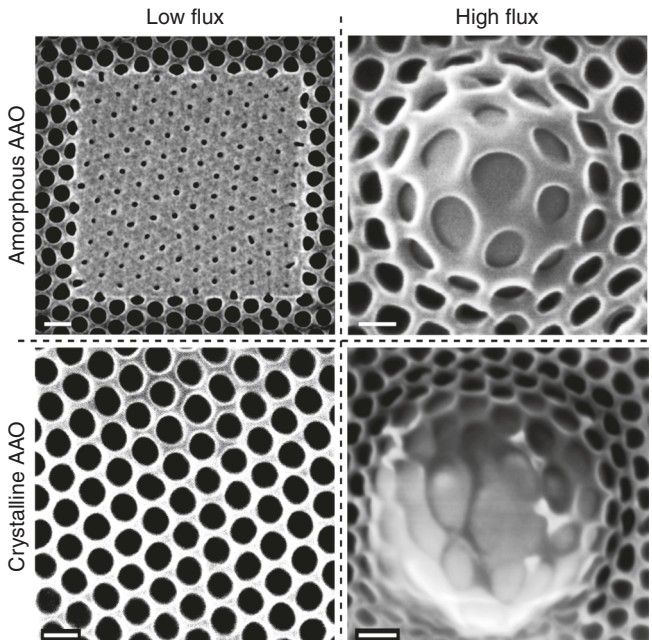

**Fig. 2** The role of ion flux and chemical bonds. The key experimental observations in amorphous and crystalline AAO under low-flux and high-flux conditions are summarized (materials: am-AAO vs c-AAO; ion beams: 'low-flux' vs 'high-flux'). Two significantly different flux-dependent regimes are observed: within the 'low-flux' regime pores in am-AAO shrink to smaller pores, while c-AAO shows remarkable resistance to irradiation; within the 'high-flux' regime (and at high fluence where the substrate swelling occurs) pores in am-AAO do not shrink anymore; however the thin film shows surprising superplastic behavior, in contrast with c-AAO, which exhibits bond breaking and irregular deformation.(The flux of $F < 10$ and $F > 100$ ions $nm^{-2} s^{-1}$, referred herein as 'low-flux' and 'high-flux'regimes, respectively). The scale bar is 100 nm

nm solid-state nanopores can be achieved using this irradiation technique (Supplementary Fig. 1). To understand the underlying mechanism of pore closure at the low-flux regime, TEM lamellas of the irradiated areas on the sample were prepared. TEM images of the pores of am-AAO at different stages of irradiation are shown in Fig. 3. Interestingly, the actual pore closure occurred deeper in the material and at later stages all the spacing between the pore columns were filled. Chemical analysis, including EELS (in TEM) as well as EDX (in SEM), suggests that the filled area of the pores is composed of two major elements: Aluminum and Oxygen (Fig. 4a–e and Supplementary Fig. 2). Additionally, the high-resolution TEM images highlight crystallization of am-AAO at both pre-existing and closed-pore regions upon irradiation with low-energy $He^+$ ions (Fig. 4f–h and Supplementary Fig. 3).

In addition to the ex situ characterization of the irradiated material at atomic scale, nanoscale dynamics of pore closure in am-AAO was monitored by in situ visualization of the nanopores during the irradiation, using the secondary electron feedback. In a typical experiment ion beam was scanned over a $100 \times 100$ nm$^2$ sample area in a $512 \times 512$ windows frame, using 0.1 µsec dwell time with arbitrary number of frames and without averaging. The flux of ions was controlled by varying currents via changing the gas pressure in the ion source with 0.1 pA resolution. The rate of pore closure for individual pores at different irradiation regimes was used as a measure for efficiency of matter transport (Fig. 5a–e). Flux-dependent nanoscopic evolution of am-AAO is plotted in Fig. 5c. At low-flux regime ($F < 10$ ions $nm^{-2} s^{-1}$) pore closure occurred very fast, while in the intermediate regime

($F = 50$ ions $nm^{-2} s^{-1}$) the pores of am-AAO shrank with a slower rate, whereas a higher fluence of ions was required to completely close a pore. At high-flux regime ($F > 100$ ions $nm^{-2} s^{-1}$), am-AAO did not show any microscopically observable decrease in pore size up to the fluences relevant to the low-flux or moderate-flux regimes ($<5 \times 10^{16}$ ions $cm^{-2}$). At higher fluences ($>1 \times 10^{17}$ ions $cm^{-2}$), the pores were slightly enlarged due to the volume swelling of the (silicon) substrate. Silicon substrate swelling due to $He^+$ irradiation occurs due to amorphization (reduced density) of the material[14]. These flux-dependent results are phenomenologically independent of scanning parameters (such as scan rate, size, or number); however, fastest scanning rate (i.e., shortest dwell-time) was implemented deliberately to achieve more homogeneous and isotropic mass-flow patterns.

In addition to irradiation flux, pore closure rate (slope of the linear fit in size vs time graph) exhibits strong dependency on ion-beam energy, incident angle, and substrate temperature. Similar flux-dependency was observed for experiments done with different ion beam energies (25–45 keV) and under different incident angles (0–85°, measured with respective to the normal to the surface). Within the low-flux regime, increasing the incident angle reduced the closure rate (Fig. 5d). At very high angle of incidence (~75–85°) no significant pore closure was observed at the relevant flux and dose (Supplementary Fig. 4). The highest closure rate corresponded to the irradiation with 25 keV ion beams at 0°. By increasing the ion-beam energy (to 30 and 45 keV), the closure rate reduced significantly (Fig. 5e). Experiments were mainly performed at room temperature, yet a designed experiment at higher temperature shows that the pore closure efficiency reduced by increasing the temperature.

Based on these observations, it is concluded that the impact of the $He^+$ on am-AAO is significant both in atomic-scales and meso-scales. The pore closure by $He^+$ ion irradiation is rather a surprising and yet complicated phenomenon. In general, the pore closure can be a convolution of multiple phenomena and processes, including sputtering[15–18], cavity swelling[19], radiation-induced diffusion[20], and a phase change[2,21]. Determining the contribution of each phenomenon is not trivial and requires more fundamental studies and comprehensive analysis approaches on each separate effect. Here we qualitatively evaluate the significance of some of the important effects in our experiments.

Sputtering is a well-known effect in Focused-Ion-Beam (FIB) process and can be a source of mass flow in irradiated materials[22,23]. Sputtering as a possible mechanism for pore closure was evaluated using Monte Carlo simulations using the software SRIM[24]. The sputtering yield of 0.10 atoms per incident ion at 0° incidence and 2.13 atoms per incident ion at 85° incidence (which corresponds to the glancing angle of the ion beam on the side walls of the pores) was determined and the amount of sputtered atoms was evaluated for the used dose of ~20 ions $nm^{-2}$. Even for glancing angles, where sputtering is maximal, the amount of sputtered atoms only corresponds to 10% of the pore volume and is not sufficient to explain pore closure. Additionally, as it is shown in Fig. 5d, sputtering yield increases with incident angle of the ion-beam in contrast with the trend of the experimental observations, in which the mass transport efficiency was reduced with incident angle.

Backsputtering of the silicon into the pores can be neglected with a sputtering yield of 0.06 silicon atoms per incident ion. This is in good agreement with further experiments which were performed on free-standing AAO membranes with no substrate (Supplementary Fig. 5). Pore closure was observed on the free-standing membrane at low-flux regime, also suggesting that the substrate is not contributing dramatically in the closure of the pores.

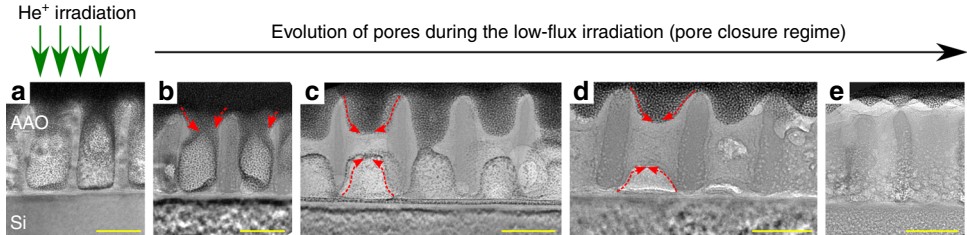

**Fig. 3** Evolution of pores during the closure. TEM images show FIB-prepared cross-sections at different fluence stages of pore closure, after low-flux irradiation of am-AAO sample on Si substrate with 25 keV He$^+$. Irradiation fluence from (**a-e**): 0, ~5 × 10$^{14}$, ~2 × 10$^{15}$, ~3 × 10$^{15}$, and ~2 × 10$^{16}$ ions cm$^{-2}$. The red arrows follow the mass-flow pattern. The scale bar is 50 nm

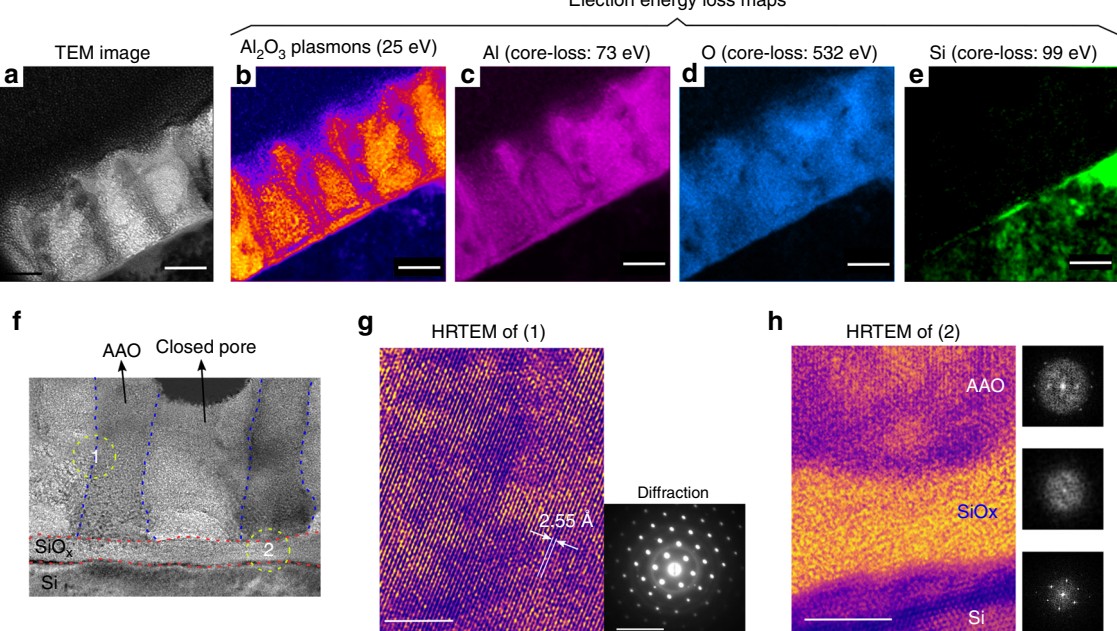

**Fig. 4** Atomic structure. **a–e** TEM image and EELS (electron energy loss spectroscopy) from a cross-section of closed pores in am-AAO irradiated with low-flux 25 keV He$^+$. The corresponding EELS maps are: plasmon-loss of Al$_2$O$_3$ at 25 eV, Al core-loss (in Al$_2$O$_3$) at 73 eV, Oxygen core-loss at 532 eV, and Si core-loss at 99 eV. It is concluded that the re-distributed material during the pore closure contains the same chemical components as the starting material (i.e.,: Aluminum and Oxygen with major Al-O bonds in the structure). The scale bar is 50 nm. **f–h** HRTEM (high-resolution TEM) images from a closed pore revealing crystallization am-AAO after irradiation, where flow of low-energy He$^+$ ions through thin am-AAO films left behind increased order in the crystal structure at both pre-existing and closed-pore regions. Dashed lines in (**f**) are to approximately indicate the boundaries of pre-existing and newly-formed (closed pore) areas. **h** Lattice fringes in the crystalized am-AAO is obvious in Region 1 in both pre-existing and closed-pore areas (the scale bar is 5 nm). The inset shows selected area electron diffraction (the scale bar is 10 1/nm). **h** Region 2, is the atomic structure at the interface of the materials (Si substrate, natural SiO$_x$ oxide layer, and AAO superstrate), in which: lattice fringes in Si substrate are (100) planes; SiO$_x$ layer remains amorphous after irradiation; AAO shows lattice fringes after irradiation (also see FFT images of each area). The scale bar is 10 nm. To see the high-resolution images please refer to Supplementary Fig. 3

Elemental analysis in TEM (EELS maps in Fig. 4), suggest that the material inside of the pores is a composition of aluminum and oxygen (possibly in a crystalline form). Hence, it is reasonable to assume that carbon deposition (another common effect in FIB[25]) is not the main source of mass flow during the irradiation with He$^+$. Particularly, the samples were plasma cleaned extensively before the irradiation experiment; therefore, low-carbon contamination is anticipated. Furthermore, pore closure was not observed for c-AAO, making carbon deposition a very unlikely candidate.

He$^+$ bubble formation inside the thin film and cavity swelling can be the other contributing sources for pore closure. It was observed that cavity swelling is more significant in moderate-flux regime compared to low-flux and high-flux regimes (Supplementary Fig. 6). Also, the thickness of the membranes increased during moderate-flux irradiation (due to cavity swelling), while it

was slightly decreased during low-flux irradiation. Therefore, it is anticipated that cavity swelling alone cannot be responsible for pore closure.

It is worth pointing that some of the mentioned effects, such as sputtering and cavity swelling scale with temperature[26–28]. However, experiments performed at higher temperatures shows that the pore closure efficiency dramatically reduced by increasing the temperature, most probably due to the reduced mean-free path of the diffusive entities at higher temperatures[7]. Local temperature increase[29] at the interaction spot of the ion beam is not expected to be significant within the keV light-ion irradiation regime[3,7]. This is also evident from our experiments, where pore closure rates reduced with increasing the beam-flux, or increasing the substrate temperature.

These observations suggest that "irradiation-induced diffusion"[30] is likely to be the most dominant source of mass-flow in

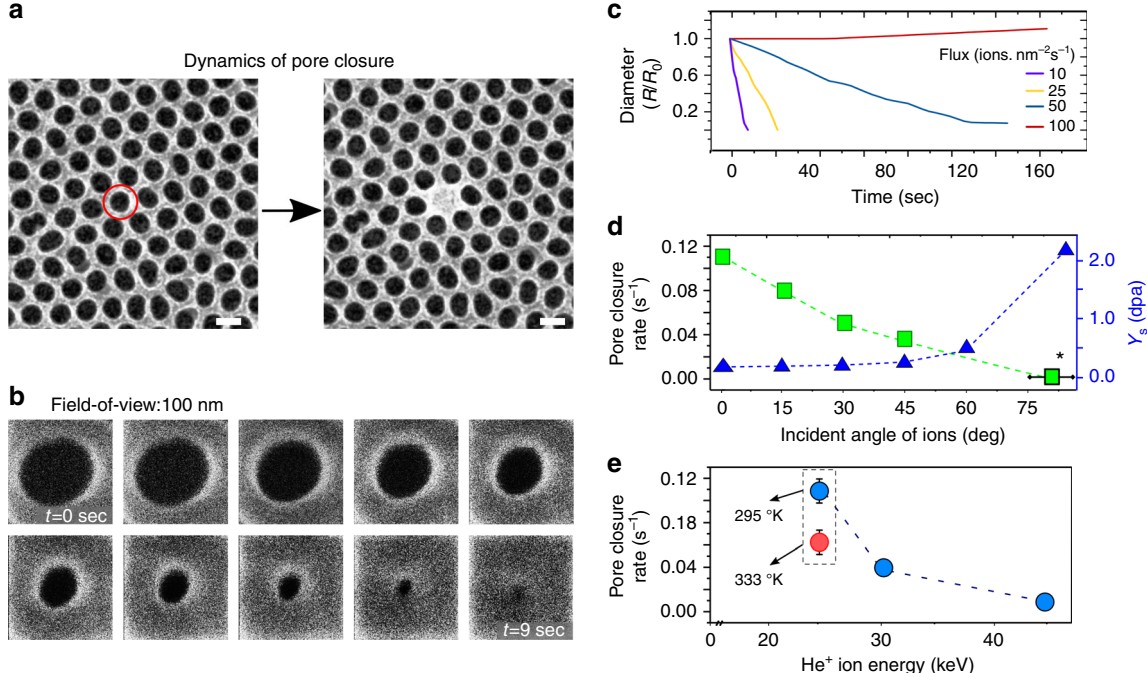

**Fig. 5** Dynamics of pore closure. **a**, **b** HIM images of am-AAO in low-flux irradiation regime at 25 keV He$^+$. **a** Shows the initial non-irradiated pore and the same pore after irradiation and closure. The scale bars are 100 nm. **b** Shows the temporal evolution of a pore (field of view of 100 nm). **c** Change in pore diameter with time (fluence) in dependence of the ion flux. The pores close faster (also at less fluence) for lower ion fluxes (10 ions nm$^{-2}$ s$^{-1}$ and 25 ions nm$^{-2}$ s$^{-1}$). A significantly slower pore closure is exhibited when using medium ion fluxes (50 ions nm$^{-2}$ s$^{-1}$). The pore diameter increases slightly for high flux irradiation (100 ions nm$^{-2}$ s$^{-1}$). **d** Pore closure rates (light squares) and calculated sputtering yield (dark triangles) as a function of incident angle of ion beam. Closure rate reduces with incident angle while sputtering yield increases. Experiments were performed on am-AAO on Si substrate; however, the measurement for 80 ± 5° (indicated by a star) was performed on a free-standing film (see SI). **e** Ion-energy dependence of pore closure rate. Increasing the temperature reduced the closure rate at 25 keV. (n.b. pore closure rate (sec$^{-1}$) is an estimate from the (negative) slope of diameter vs time graph, assuming a linear behavior)

pore closure. Indeed, few other studies have shown that low-energy ion beam irradiation results in lateral mass-flow in thin film nanomaterials due to irradiation-induced diffusion/flow[6,8,22]. Irradiation-induced VLS growth of nanowires (vapor-liquid-solid) using keV He$^+$ beam is another evidence for the significance of diffusion in the relevant experiments.[4] Generally, atoms in a low-dimensional materials subjected to irradiation experience axial stress[31,32], in which the relaxation can be accompanied by atomic diffusion/flow[30]. Injection of 'adatoms'—e.g., interstitials and vacancies by ion beam (and also other 'defects')—can promote the irradiation-induced diffusion[33]. We suggest that irradiation-induced ionization of the material is a critical factor in diffusion of these mobile entities.

The concentration of the ion-induced 'adatoms' is proportional to irradiation flux[34]; however, mobility of these entities depend on other parameters such as recombination rates[11], electron's mean-free path[10], structural defects[12], chemical bonds[10], stress[34], temperature[7], and electrostatic forces[8]. Generally, enhancement in mobility/diffusion of these entities leads to improved mass-flow rates, and, therefore, efficiency in pore closure. In materials with high ionicity, such as alumina, the amount of the irradiation-induced ionization determines the mean-free-path of the defects[11]. At high ion-beam fluxes, higher ion-induced ionization reduces (and eventually freezes) the mobility of the defects, implying slower rate of lateral mass-flow and subsequent mesoscopic morphological changes. This is in-line with the observed flux-dependent pore closure at different ion energies.

Ionization of material under irradiation, not only influences the mobility of defects, but also constrains the paths for ionic mass-flow in both short and long-range distances. In short-range distance, irradiation-induced diffusion promotes formation of nanocrystalline Al$_2$O$_3$ structure (Fig. 4f–h). Crystallization was not observed in the SiO$_x$ layer (natural oxide) in the Si substrate, most probably due to the fact that Si-O bonds are covalent bonds and they cannot form ionized networks in their amorphous structure.

Influence of ionization in long-range distance can be possibly observed in the patterns of mass-flow during the pore closure (Fig. 3). Finite-element-method simulations suggest that the ionic mass-flow of O$^{-2}$ and Al$^{+3}$ in a homogenously ionized structure produces anisotropic flow patterns with accumulation tendency at the central region of pore walls (Supplementary Fig. 7).

c-AAO—obtained by annealing of AAO at higher temperatures—has almost identical microscopic morphologies as am-AAO, but has a different atomic structure and chemical bonds[35]. The pre-annealing of AAO results formation of different phases of stoichiometric crystalline Al$_2$O$_3$ with majority of ionic bonds, but also it substantially reduces the defects and impurities concentration due to segregation and volatilization[35]. Radiation resistance in c-AAO—which has much higher ionic bonds compared to am-AAO—most probably comes from profound irradiation-induced ionization and long-range charge ordering and inter-atomic correlations[36]. In such a highly ionized crystal, charged ions are strongly correlated and it would require additional energy to displace an atom in this network. Therefore, the charge-rich correlated network of ions creates a barrier for atomic diffusion, which makes c-AAO resistant to irradiation (both at low-flux and high-flux regimes). Other studies, with different ions and sources,

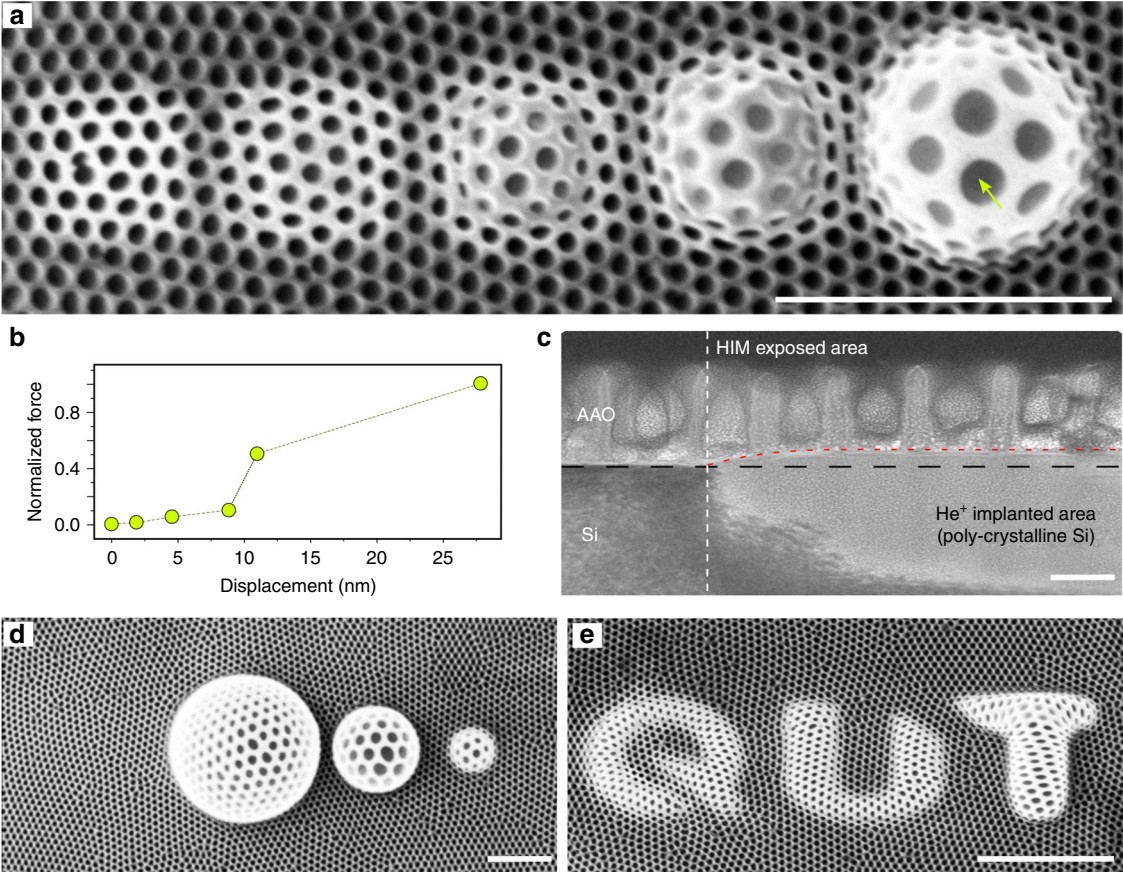

**Fig. 6** Superplastic shaping of AAO. **a** Ion-induced damages in Si produced patterns on the surface, which forcefully stretched and bended the superstate am-AAO thin film during the irradiation. am-AAO elastically/plastically responded to the morphological changes of the substrate in both nanometer and micrometer scales. The irradiation fluence from left to right: $1 \times 10^{16}$, $5 \times 10^{16}$, $1.0 \times 10^{17}$, $5.0 \times 10^{17}$, and $1.00 \times 10^{18}$ ions cm$^{-2}$. **b** Inelastic deformation of am-AAO in response to the imposed forces from the substrate. **c** TEM cross-section of the implanted area at the high-flux regime (and the comparison to non-irradiated area). The volume increase due to amorphization of Si substrate (fluence: $1 \times 10^{16}$ ions cm$^{-2}$) is evident by the difference in contrast between the two regions. The scale bar is 100 nm in the TEM image. **d** shows the possible extend of helium ion irradiated am-AAO superplastic stretching with $1.00 \times 10^{18}$ ions cm$^{-2}$ at 3 different scales. **e** Shows that different nanoscale patterns can be produced by He$^+$ implantation in the substrate at the high-flux regime. The scale bar is 1 μm in HIM images

also have shown that alumina is extremely resistant to ionizing irradiation that has found applications in reactor coatings and swift heavy-ion-beam lithography[37,38]. Ion-channeling in crystalline materials is another source of resistivity against damage in irradiation experiments; however, since irradiation resistivity of c-AAO is independent of the incident angle of the ion beam, it is reasonable to assume that ion-channeling is not the major influential parameter in irradiation resistivity of c-AAO.

**Superplastic-like behavior**. Generally, nanoporous am-AAO thin films are brittle and exhibit very little inelastic deformation under tensile stresses (Young's modulus of ~30–100 GPa)[39–41]. It is known that thin films of am-AAO are not able to sustain large plastic flow at room temperature[42]. After fabrication of thin membranes, any mechanical re-shaping of these membranes is practically impossible. For example, in a designed experiment, it was observed that at relatively low values of strain (up to ~5%), 100-nm-thick am-AAO fails with significant microscopic cracks and ruptures on the structure. (The same behavior was observed when the thin films were slightly bended—please refer to Supplementary Fig. 8).

A surprisingly different behavior was observed when the am-AAO thin films were irradiated with He$^+$ ion beam within the

high-flux regime (Fig. 6). At the high irradiation fluence, where the silicon substrate amorphization and volumetric swelling occurs, am-AAO exhibited anomalous plastic deformation in response to the imposed creep. Figure 6a shows the episodes of uniform deformation of the nanoporous membrane at different stages of expansion due to the swelling substrate (Fig. 6c). The corresponding material elongation (i.e., pore size vs normalized stress) is plotted in Fig. 6b, showing inelastic response of the material to substrate forces. Large uniform enlargement of nanopores could be observed at high tensile stresses, where the material exhibited superplasticity. For example, a pore (indicated with an arrow in Fig. 6a) with an initial diameter of 78 nm was enlarged to 197 nm after deformation (252% elongation). This uniform deformation was not observed for c-AAO, though.

Unprecedented opportunities may arise when normally brittle nanoporous structures could plastically deform after their fabrication into structures and devices. Figure 6d, e show some examples of ion patterning of the substrate, in which the local volumetric swelling of the substrate was accompanied by superstrate bending and deformation to create novel structures which were not possible to obtain before.

Plastic flow in am-AAO is continuous and smooth without any significant rupture in the microscopic structure. The swelling

substrate ruptures earlier than the am-AAO film, making it impractical to test the failure of the am-AAO under extreme tensions (see Supplementary Fig. 9). It should be mentioned that the deformation of the Si substrate with He$^+$ irradiation is also superplastic and it has been reported previously by Livengood et al.[14]. Moreover, generally the ion beam (or electron-beam) facilitated plastic flow in nanomaterials has been discussed in some reports[3,34,43]. Our system of study allows studying the role of the nature of chemical bonds and disorder in irradiation-induced plastic deformations. It should be noted that the observed superplastic-like behavior is induced by ion irradiation at room temperature and might be of a different nature compared to the superplasticity commonly attributed to grain boundary sliding in matter subjected to high temperatures[44,45]. Also the observed phenomena is of a different nature compared to the enhanced ductility of materials due to irradiation-induced bubble formation[46,47].

To account for observed superplastic behavior of am-AAO, one should consider the chemical structure of the material and the role of impurities and defects in observed plastic behavior[10,33,37]. A general rule is that materials exhibit plasticity if the chemical bonding allows accommodation of imposed deformations[43,48]. am-AAO—unlike c-AAO—contains large number of defects and impurities in its structure[13,41,49]. The intrinsic impurities (such as -OH$_2{}^+$ and -CO$^-$) majorly appear during the field-assisted oxidation of aluminum in acidic solutions, which are highly reduced after annealing at high temperatures. It is foreseen that the impurities and defects in materials can promote irradiation-induced deformations[50]. Existence of chemical disorder in atomic structure can substantially reduce the thermal conductivity, due to reduced mean-free-path of electrons, and subsequently reduce the energy dissipation rates[48].

Energy dissipation in ionic materials is relatively fast and the energy cost for bond breaking is too high and eventually deleterious. As a result, irradiation of highly ionic materials, such as c-AAO, may not allow the material to sustain shear deformation by chemical bond switching and re-structuring[43,48,50]. On the other hand, covalent bonds are more flexible to atomic re-arrangement under irradiation because they are only constrained by their nearest atomic bond[51]. The chemical bonding of impurities in am-AAO majorly have covalent nature, and, therefore, they can produce local defects in the material, which are then able to disrupt the local packing of ionic bonds and influence the mean-free-paths of electrons/ions and energy dissipation channels of neighboring atoms. One possibility is that these defects can act as bridging sites to overcome the energy barrier for bond switching, which in fact would mediate the oxide viscosity. As a results, chemical bonding in covalent materials can allow accommodation of imposed deformations[43,48], which can explain the plasticity of am-AAO under irradiation.

It is concluded that nanoscopic ion-matter interactions can be effectively manipulated by ion-beam flux and chemical structure of the target material. Two distinct interaction regimes were found when irradiating nanoporous am-AAO with low-energy helium ions: (i) Lateral mass-flow and pore closure at the low-flux regime; (ii) Superplasticity at the high-flux regime. By studying different parameters, such as irradiation flux and the nature of chemical bonds in the target materials, it is concluded that whilst diffusive entities (adatoms) tend to flow in the material upon irradiation-induced effects, irradiation-induced ionization tends to create potential barriers for mass-flow. Dominance of these competing mechanisms is determined by the ion-beam flux and the relative presence of ionic and covalent chemical bonds in the material. The existence of chemical impurities in an ionic network is necessary to enhance ion-induced mass-flow and

plasticity, confirmed through the comparison of am-AAO and c-AAO (Fig. 1).

The results suggest that the ion-matter interactions at atomic scales and the arising mass transport over nanoscopic to microscopic scales can be controlled during the irradiation, enabling potential deterministic shaping of nanomaterials. The implemented approach positively answers the question posed in the Introduction as the atomic structure allows control over nanoscale, and microscale morphology of the materials under irradiation. Indeed, large scale arrays of ultra-high-density nanopores (sub-10 nm) were fabricated by shrinking the pores of am-AAO. Moreover, extruding and bending of am-AAO thin films enabled fabrication of structures with complex architecture. The mesoscopic deformation of am-AAO can produce shapes and structures, which were not possible to obtain with other fabrication techniques. The nanoscale resolution of the micro-scope allows high-precision patterning of the substrate which is then imposes the corresponding geometrical evolution to the superstrate. The discovered possibilities may thus open numerous opportunities for the multi-scale tuning of the structure, properties and performance of diverse materials through ion beam-induced manipulation of atomic bonds and structural disorder. More fundamental studies are required to determine the irradiation-induced diffusion length scales and compare them with the physical length-scale of nanomaterials.

**Data availability**. All data needed to evaluate the conclusions in the paper are present in the paper. Additional data related to this paper may be requested from the authors.

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

## Acknowledgements

The authors would like to greatly acknowledge contribution of Dr. Peter Hines at (CARF) QUT in maintenance of the HIM microscope and also fruitful discussions. We acknowledge research facility and technical assistance from the CARF (Central Analytical Research Facility at Queensland University of Technology), specially Dr. Jamie Riches. Authors thank the technical assistance from Dr. Graeme Auchterlonie at microscopic facility of the University of Queensland for EELS and EDX analysis. The authors acknowledge the facilities (FEI Scios) at the Australian Microscopy & Microanalysis Research Facility at the Centre for Microscopy and Microanalysis, The University of Queensland. We thank Prof. Robert G. Elliman at the Research School of Physics and Engineering at Australian National University, for fruitful discussions and also for performing the RBS experiments and analysis. We thank Dr. Sergey Rubanov at the University of Melbourne, Dr. Ivan Shorubalko at the Swiss Federal Laboratories for Materials Science and Technology (EMPA) and Prof. Janos Vörös at ETH Zurich for fruitful discussions. KO acknowledges partial support by the Australian Research Council and CSIRO-QUT Joint Sustainable Processes and Devices Laboratory. MA acknowledges Marie Skłodowska-Curie actions (Project Reference: 706930).

## Author contributions

M.A. developed the concept and designed the experiments. Y.M. delivered AAO fabrication and transfer technology. M.A and A.W. carried out the experimental investigations. A.W. carried out SRIM/TRIM investigations. K.O. contributed to concept advancement, data analysis, manuscript preparation, and revision. M.A. wrote the manuscript through the collaboration of all the authors.

## Additional information

**Competing interests:** The authors declare no competing interests.

