## [Peer Review File · Nature Communications]

Reviewers' comments:

Reviewer #1 (Remarks to the Author):

This paper studies the restructuring and deformation upon He induced volumetric swelling and He ion beam implantation. The paper contains two parts a) closure of nanopores due to He ion impact and b) deformation of the alumina film by swelling.

While this paper does show interesting and very nice figures and phenomena I do not think it is suitable for this journal due to the fact that a number of claims made are not supported by the data provided and the conclusions may not hold. Especially the pore closure part seems weak.

Major points are as follows:

1) The Authors do not provide any detail on sputtering effects. The authors simply state there is minimal sputtering. You need to show that some of the pore closure does not originate from redeposited sputtered material. It is key to consider glancing angle sputtering as well as heads on sputtering. No data of sputter rate as a function of impact angle and ion energy is provided

2) The authors state that the film experiences plastic flow. Why can it not be stretched elastically? The elastic modulus of such a nanoporous structure is not that high. This is a very thin porous thin film and thin films can deform significantly elastically as well. Is there any data on tensile testing on Alumina nanoporous films in the literature supporting the claim of superplastic behavior on films with this thickness? To what strain can one take a sample before the film breaks and crack might be observed? Maybe the FEM can provide some insight.

3) There is extensive discussion on nuclear stopping vs. electronic stopping in the supporting documents. But detailed parameters from the SRIM calculations are missing. What displacement energy was used? What Cal caution mode was used? Why is it important how the ions lose the energy while the simple fact of displaced ions might be sufficient to explain enhanced diffusion. It seems this discussion on this point is more complicated than it needs to be.

4) In general the observed phenomena are a convolution of so many phenomena and processes that it is difficult to explain them so simply here. I would argue the pore closure is a combination of sputtering, cavity swelling, radiation induced diffusion and a phase change. How each phenomena contributes is very difficult to say without a more fundamental separate effects testing and analysis approach.

Further minor points are below:

Figure S2 a-c: The reader cannot really see a difference between the figures. Higher resolution images would be preferred. In the document figure a b and c look very similar to the eye. It would be better to show a 5 μ m x 5 μ m field of view or smaller with clear labeling of the pore diameter.

Figure s4-2: You are making the point that the material switches between crystalline and amorphous. It would help the reader to show an insert of a FFT to foster the point of crystallization upon ion beam irradiation. Also, always show the diffraction image. This should be significantly clearer and one can see at what orientation the image was taken.

Figure S5: it is not clear to me how I am supposed to see these three mechanism in this figure. I am not sure how I am supposed to see in this figure that defects are annihilated at boundaries. Also, the figure caption should state the irradiation condition. How do you distinguish between voids and bubbles? If you cannot prove it is bubbles vs. voids please call them cavities. In general the figure caption should contain a figure description now a discussion point.

Figure S7: Please don't show just screen shots of the SRIM calculations. You can export the calculations as txt files and replot them in a graphical software. I do not understand figure 7a and the point of table b.

Figure S10-2 why is time an important parameter should this not be #ions? I can't see how the time is important here.

You are stating "The images clearly show that the displacement of atoms was perpendicular to the direction of the impinging ions." This is incorrect. All this figure shows is the fact that the material swells perpendicular to the ion beam penetration direction. This is not the same as the displacement of atoms in that direction. The swelling can of course only occur in the direction where there is space to swell into. This has little to do with the displacements of the atoms from the collision.

The manuscript does not talk about sputtering effects besides stating it is low. You need to state what it is on the heads on collision and the glancing angle collision. You need to state what the total amount of sputtered material is as a function of ions and ion energy's in the system. Can it not simply be that material is sputtered from the surface and redeposited? Also the side walls of the nano pores are seeing a glancing angle but the SRIM calculations and arguments are made as a heads on collision. This deserves more attentions.

This statement seems odd: " Additionally, a close-up view at the TEM images highlights molecular transformation" why say molecular transformation and not simple "crystallization".

What does the 0.28 and 0.6 displacements/A/ion relate to in dpa at the different doses? The percentage of displaced atoms is a strange unit to use.

It seems odd that the manuscript and figure S1 only talks about the HIM as an ion beam irradiation source and then later suddenly some RBS with 400keV is mentioned. What were the details of the RBS current, etc..

There seems some statement about a "force field" as the reason why the pores close in the middle rather than elsewhere. I really don't know what force field this might be and this statement " The force field is most probably related to the ionization of the material and long-range electrostatic forces." Does not help. What electrostatic forces?

How can you say the center contains more mass? I do not see an estimate of mass. Can it not just be that you have material of lower density there due to nanocavities formed? Several papers show that one forms 2-5nm sized cavities in materials upon HIM or general room temperature He implantation. Can you do an estimate of bubble size vs. volumetric changes? How much of the volumetric changes can be attributed to bubbles and what has to be a different mechanism like voids or diffusion?

I do not know what physical materials property "stretchability" is? Can you relate this to a physical property such as elongation until fracture or similar? Is there any data on applying a stress to these nanoporous thin films you can compare this too? I am sure someone did a tensile test on these films and evaluated at what strain the film ruptures.

This is simply wrong" Generally plastic deformation is not observed in crystal structures.." are you saying no crystal structure can be plastically deformed? What about metals?

Reviewer #2 (Remarks to the Author):

The paper deals with understanding the interaction of low-energy ion beams (mainly 25kV Helium ions) with nanoporous alumina supported by silicon substrate. The major claims of the paper are:

- identification of two distinct regimes: (i) lateral mass-flow and pore closure at the low-flux regime, (ii) superplasticity at the high-flux regime.
- potential bridging of adatom diffusion and viscous models
- the implemented approach allows precise control over the atomic structure, nanoscale, and microscale morphology of the materials.

Large amount of experimental data is presented in the manuscript: 9 multi-panel figures in the main text and additional 10 in the supporting information. Nevertheless, this large amount of data does not help to clearly understand the main focus of the paper.

One of the most important figures in the paper is Fig.2 – showing cross-section TEM images of the AAO sample on Si substrate. The irradiation is done in the “low-flux” regime and figures from (a) to (e) supposed to show effect of increasing irradiation dose. Nevertheless, the dose is not increasing homogeneously (0, 4.6×10^{14} , 1.84×10^{15} , 2.76×10^{15} , and 2.30×10^{15} ions/cm²). Authors argue about change in the thickness of AAO layer, decrease from (a) to (b) and then increase from (b) to (e). As a reader I understand that the images are taken from the different parts of the sample and that the thickness variation of a few nm would influence a lot the conclusions about layer thickness decrease from (a) ~170-180nm according to the image to (b) ~160-170nm. In any case it could be expected from the sputtering by FIB. Have authors estimated the sputtering yield of AAO by He-FIB? Image (c) shows then an increase of thickness of the AAO layer to ~180-190nm and in addition bridges formed in the middle of the pores. There are no blue arrows indicating formation of bubbles on this image. The volume of AAO layer increased substantially, what is the reason for this? Red arrows indicating lateral mass-flow do not explain the increase of the volume. Converting the material from amorphous to crystalline (as indicated in Fig3) would also not explain almost factor of 2 volume growth.

Two well-known, but not mentioned in the paper, effects that should be considered here are:

- Redeposition of the sputtered material, especially Si substrate (supported by EDX map Fig.S3). AAO layer is not conductive and charging under FIB can lead to a complicated surface electric field profiles that influence substantially the redeposition process. Redeposition effects are reported for example in Ref. D. A. M. de Winter et al, J. Vac. Sci. Technol. B 25, 2215 (2007) "Redeposition characteristics of focused ion beam milling for nanofabrication", L.A. Giannuzzi et al, Micron 30 (1999) 197–204, "A review of focused ion beam milling techniques for TEM specimen preparation".

- Carbon deposition changes substantially the shape of the FIB exposed regions. It is also known that at low-flux conditions the carbon deposition speed is limited by the FIB flux, and at high-flux conditions it is limited by available carbon (hydrocarbon) concentration around the incident point. This is consistent with the main observation of the paper - shrinkage of the pores at low-flux and slight etching at high-flux (fig.5). Carbon deposition is well explained for example in Ref. N. Kalhor, et al, Microelectronic Engineering 114 (2014) 70–77 "Sub-10 nm patterning by focused He-ion beam milling for fabrication of downscaled graphene nano devices".

Authors relate observed effects to the nature of chemical bonds. It should be mentioned that all interactions of FIBs with matter are described taking into account the strength of chemical bonds and the so called “displacement energy”. Actually, one of the ways to modify chemical bonds during FIB sputtering of the matter is introducing special chemical gasses in the vicinity of the etching or illuminating by light. See for example: L. R. Harriott, Jpn. J. Appl. Phys. 33 (1994) 7094, "Focused-ion-beam-induced gas etching", M. G. Stanford, et al, ACS Appl. Mater. Interfaces 2016, 8,

29155–29162, "Laser-Assisted Focused He⁺ Ion Beam Induced Etching with and without XeF₂ Gas Assist".

Authors state the discovery of the extraordinary superplasticity of the normally brittle porous alumina under specific ion irradiation. The effect is not unexpected and has already been discovered in a number of other material systems. For example in Ref. [R. Livengood, et al, J. Vac. Sci. Technol. B 27, 3244 (2009), "Subsurface damage from helium ions as a function of dose, beam energy, and dose rate"] it is reported that 220nm thick Si-membrane can be bent to the extreme curvatures by He-FIB irradiation, Fig. 7(b) in the reference. The role of generated by the irradiation defects in the superplasticity effect should be addressed.

Authors state that the implemented approach allows precise control over the atomic structure, nanoscale, and microscale morphology of the materials. This reviewer sees a trend in structural change, but "precise control" is a strong overestimate.

To summarize, this referee can neither conclude, nor exclude the proposed explanations and main conclusions of the paper from the presented data. It may be helpful to make an experiment on freestanding AAO membranes of different thicknesses and under control of low C-deposition. This referee does not recommend the manuscript in present form for publication in the Nature Communications.

Highlighted Remarks:

Briefly, the new data in the manuscript contains all the key experiments requested by both referees, including (but not limited to):

1. Experiments/Simulations performed at different irradiation angles including the *glancing angle*. As it was suggested by Reviewer#1, these results are able to rule out the significance of the sputtering effects.
2. Experiments performed on free-standing membranes to address the redistribution from the substrate (suggested by Reviewer#2).
3. Experiments performed at higher temperature to address contribution of temperature-dependent effects.
4. Revisited experimental data (and relevant discussions) on elemental analysis to rule out carbon deposition.
5. Experiments performed on the elastic properties of non-irradiated thin films. Plus the significance of the superplasticity of the irradiated material is highlighted with additional data.

Reviewer #1 (Remarks to the Author):

“This paper studies the restructuring and deformation upon He induced volumetric swelling and He ion beam implantation. The paper contains two parts a) closure of nanopores due to He ion impact and b) deformation of the alumina film by swelling.

While this paper does show interesting and very nice figures and phenomena I do not think it is suitable for this journal due to the fact that a number of claims made are not supported by the data provided and the conclusions may not hold. Especially the pore closure part seems weak.”

The paper is re-written. New figures and experimental results are added to support the claims of the new manuscript. Particularly, extensive discussion is dedicated for the “pore closure” section, pages 3-6 of the manuscript. We thank the referee for reading the manuscript in great details and for the valuable constructive comments.

Major points are as follows:

“1) The Authors do not provide any detail on sputtering effects. The authors simply state there is minimal sputtering. You need to show that some of the pore closure does not originate from redeposited sputtered material. It is key to consider glancing angle sputtering as well as heads on sputtering. No data of sputter rate as a function of impact angle and ion energy is provided”

The sputtering effects are extensively considered in the current manuscript by including relevant experimental/simulation results, such as dependence on angle of incidence. Particularly, experiments performed at glancing angle corroborate our claims that the sputtering is not the main source for pore closure. (Page 4 and 5 of the manuscript, **Figure 4** and **Figure S4**).

“2) The authors state that the film experiences plastic flow. Why can it not be stretched elastically? The elastic modulus of such a nanoporous structure is not that high. This is a very thin porous thin film and thin films can deform significantly elastically as well. Is there any data on tensile testing on Alumina nanoporous films in the literature supporting the claim of superplastic behavior on films with this thickness? To what strain can one take a sample before the film breaks and crack might be observed? Maybe the FEM can provide some insight.”

The “superplasticity” part is now re-written (pages 7 and 8). New experimental results are included regarding the tensile testing of the nanoporous alumina films (**Figure 5** and **Figure S8**). Relevant articles on elastic properties of nanoporous alumina films are now cited in the main text. The new experimental results and works from others confirm that normally nanoporous alumina films are brittle and exhibit very little inelastic deformation under tensile stresses. Superplastic property of these films is unusual and is reported for the first time in this report.

“3) There is extensive discussion on nuclear stopping vs. electronic stopping in the supporting documents. But detailed parameters from the SRIM calculations are missing. What displacement energy was used? What Cal caution mode was used? Why is it important how the ions loose the energy while the simple fact of displaced ions might be sufficient to explain enhanced diffusion. It seems this discussion on this point is more complicated than it needs to be.”

In the current manuscript the extensive discussion is simplified to great extends. All the details of the performed simulations are now included in the relevant part.

“4) In general the observed phenomena are a convolution of so many phenomena and processes that it is difficult to explain them so simply here. I would argue the pore closure is a combination of sputtering, cavity swelling, radiation induced diffusion and a phase change. How each phenomena contributes is very difficult to say without a more fundamental separate effects testing and analysis approach.”

This point is highlighted in paragraph 3 on page 4. The contribution of each plausible mechanism is then assessed based on the experimental results (pages 5 and 6). The conclusion is that “irradiation-induced diffusion” is the key mechanism of pore closure.

“Further minor points are below:”

“Figure S2 a-c: The reader cannot really see a difference between the figures. Higher resolution images would be preferred. In the document figure a b and c look very similar to the eye. It would be better to show a 5um x 5um field of view or smaller with clear labeling of the pore diameter.

Figure s4-2: You are making the point that the material switches between crystalline and amorphous. It would help the reader to show an insert of a FFT to foster the point of crystallization upon ion beam irradiation. Also, always show the diffraction image. This should be significantly clearer and one can see at what orientation the image was taken.”

Corrected.

“Figure S5: it is not clear to me how I am supposed to see these three mechanism in this figure. I am not sure how I am supposed to see in this figure that defects are annihilated at boundaries. Also, the figure caption should state the irradiation condition. How do you distinguish between voids and bubbles? If you cannot proof it is bubbles vs. voids please call them cavities. In general the figure caption should contain a figure description now a discussion point.”

Corrected.

“Figure S7: Please don’t show just screen shots of the SRIM calculations. You can export the calculations as txt files and replot them in a graphical software. I do not understand figure 7a and the point of table b. “

Corrected.

“Figure S10-2 why is time an important parameter should this not be #ions? I can’t see how the time is important here.”

The figure is removed from SI in the new submission.

You are stating “The images clearly show that the displacement of atoms was perpendicular to the direction of the impinging ions.” This is incorrect. All this figure shows is the fact that the material swells perpendicular to the ion beam penetration direction. This is not the same as the displacement of atoms in that direction. The swelling can of course only occur in the direction where there is space to swell into. This has little to do with the displacements of the atoms from the collision. “

Corrected. The pore closure mechanism is explored in more details in the new manuscript and different possible mechanisms are investigated. It is shown that swelling alone cannot be the dominant mechanism in pore closure. The relevant discussion is on Page 4 and 5 of the new manuscript.

“The manuscript does not talk about sputtering effects besides stating it is low. You need to state what it is on the heads on collision and the glancing angle Collision. You need to state what the total amount of sputtered material is as a function of ions and ion energy’s in the system. Can it not simply be that material is sputtered from the surface and redeposited? Also the side walls of the nano pores are seeing a glancing angle but the SRIM calculations and arguments are made as a heads on collision. This deserves more attentions.”

Corrected. We have now critically investigated the sputtering effect and confirmed that sputtering is not the dominant mechanism in our system of study. New experimental data on glancing angles and simulation results are provided in the manuscript (page 4 and 5).

“This statement seems odd:” Additionally, a close-up view at the TEM images highlights molecular transformation” why say molecular transformation and not simple “crystallization”. “

Corrected.

“What does the 0.28 and 0.6 displacements/A/ion relate to in dpa at the different doses? The percentage of displaced atoms is a strange unit to use.”

Corrected.

“It seems odd that the manuscript and figure S1 only talks about the HIM as an ion beam irradiation source and then later suddenly some RBS with 400keV is mentioned. What were the details of the RBS current, etc..”

Corrected. RBS data is not presented in the new manuscript to help with the focus of the paper.

“There seems some statement about a “force field” as the reason why the pores close in the middle rather than elsewhere. I really don’t know what force field this might be and this statement” The force field is most probably related to the ionization of the material and long-range electrostatic forces.” Does not help. What electrostatic forces?”

Corrected terminology is used to describe the ionization and electrostatic forces in the new manuscript.

“How can you say the center contains more mass? I do not see an estimate of mass. Can it not just be that you have material of lower density there due to nanocavities formed?”

Corrected.

“Several papers show that one forms 2-5nm sized cavities in materials upon HIM or general room temperature He implantation. Can you do an estimate of bubble size vs. volumetric changes? How much of the volumetric changes can be attributed to bubbles and what has to be a different mechanism like voids or diffusion?”

The related discussion is now on page 10 of the SI. Unfortunately, due to complications in the measurement we were not able to quantify the bubble size distribution. However, a relative qualitative example is shown in Figure S6.

“I do not know what physical materials property “stretchability” is? Can you relate this to a physical property such as elongation until fracture or similar? Is there any data on applying a stress to these nanoporous thin films you can compare this too? I am sure someone did a tensile test on these films and evaluated at what strain the film ruptures.”

Corrected. We have performed new experiments to measure the elasticity of thin film AAO membranes used in our work. The results and setup of the experiment are shown in Figure S8.

“This is simply wrong” Generally plastic deformation is not observed in crystal structures..” are you saying no crystal structure can be plastically deformed? What about metals?”

Corrected.

Reviewer #2 (Remarks to the Author):

“The paper deals with understanding the interaction of low-energy ion beams (mainly 25kV Helium ions) with nanoporous alumina supported by silicon substrate. The major claims of the paper are:

- identification of two distinct regimes: (i) lateral mass-flow and pore closure at the low-flux regime, (ii) superplasticity at the high-flux regime.

- potential bridging of addatom diffusion and viscous models

- the implemented approach allows precise control over the atomic structure, nanoscale, and microscale

morphology of the materials. Large amount of experimental data is presented in the manuscript: 9 multi-panel

figures in the main text and additional 10 in the supporting information. Nevertheless, this large amount of data does not help to clearly understand the main focus of the paper.”

The paper is re-written and within the light of the referees' comments from our previous submission to Nature Communications, the new manuscript tends to be more focused, precise and more dense in context. Some of the unnecessary data and discussions are excluded from the previous version to help the reader with more focused flow.

“One of the most important figures in the paper is Fig.2 – showing cross-section TEM images of the AAO sample on Si substrate. The irradiation is done in the “low-flux” regime and figures from (a) to (e) supposed to show effect of increasing irradiation dose. Nevertheless, the dose is not increasing homogeneously (0, 4.6×10^{14} , 1.84×10^{15} , 2.76×10^{15} , and 2.30×10^{15} ions/cm²). Authors argue about change in the thickness of AAO layer, decrease from (a) to (b) and then increase from (b) to (e). As a reader I understand that the images are taken from the different parts of the sample and that the thickness variation of a few nm would influence a lot the conclusions about layer thickness decrease from (a) ~170-180nm according to the image to (b) ~160-170nm. In any case it could be expected from the sputtering by FIB. Have authors estimated the sputtering yield of AAO by He-FIB? Image (c) shows then an increase of thickness of the AAO layer to ~180-190nm and in addition bridges formed in the middle of the pores. There are no blue arrows indicating formation of bubbles on this image. The volume of AAO layer increased substantially, what is the reason for this? Red arrows indicating lateral mass-flow do not explain the increase of the volume. Converting the material from amorphous to crystalline (as indicated in Fig3) would also not explain almost factor of 2 volume growth.”

Indeed the thickness was not increased in those experiments. However, we realize that due to the variation in thickness of AAO, it will not be straight forward to draw conclusions based on observation on thickness change in TEM images. Therefore, all the statements related to thickness change based on TEM images are excluded in the new manuscript. The scale bars in the image are re-plotted.

“Two well-known, but not mentioned in the paper, effects that should be considered here are:”

“- Redeposition of the sputtered material, especially Si substrate (supported by EDX map Fig.S3). AAO layer is not conductive and charging under FIB can lead to a complicated surface electric field profiles that influence substantially the redeposition process. Redeposition effects are reported for example in Ref. D. A. M. de Winter et al, J. Vac. Sci. Technol. B 25, 2215 (2007) "Redeposition characteristics of focused ion beam milling for nanofabrication", L.A. Giannuzzi et al, Micron 30 (1999) 197–204, "A review of focused ion beam milling techniques for TEM specimen preparation".”

New data based on the experiment on free-standing membranes are now included in the manuscript. The chemical analysis (EELS) is now a part of the main manuscript (Figure 3). Relevant discussions are on Page 5, paragraph 1,2,3.

A note is added to the supporting information (EDX map on Figure S2) regarding the spatial resolution of SEM EDX to avoid confusion.

“- Carbon deposition changes substantially the shape of the FIB exposed regions. It is also known that at low-flux conditions the carbon deposition speed is limited by the FIB flux, and at high-flux conditions it is limited by available carbon (hydrocarbon) concentration around the incident point. This is consistent with the main observation of the paper - shrinkage of the pores at low-flux and slight etching at high-flux (fig.5). Carbon deposition is well explained for example in Ref. N. Kalhor, et al, Microelectronic Engineering 114 (2014) 70–77 "Sub-10 nm patterning by focused He-ion beam milling for fabrication of downscaled graphene nano devices".”

This comment is also relevant to the previous one. The relevant discussion is on Page 5, paragraph 3.

Briefly, based on the experimental results in the “pore closure” section of the manuscript, the major contribution from carbon deposition or Si substrate re-depositions are ruled out.

Authors relate observed effects to the nature of chemical bonds. It should be mentioned that all interactions of FIBs with matter are described taking into account the strength of chemical bonds and the so called “displacement energy”. Actually, one of the ways to modify chemical bonds during FIB sputtering of the matter is introducing special chemical gasses in the vicinity of the etching or illuminating by light. See for example: L. R. Harriott, Jpn. J. Appl. Phys. 33 (1994) 7094, "Focused-ion-beam-induced gas etching", M. G. Stanford, et al, ACS Appl. Mater. Interfaces 2016, 8, 29155–29162, "Laser-Assisted Focused He+ Ion Beam Induced Etching with and without XeF2 Gas Assist".

All the references are cited in the new manuscript.

Authors state the discovery of the extraordinary superplasticity of the normally brittle porous alumina under specific ion irradiation. The effect is not unexpected and has already been discovered in a number of other material systems. For example in Ref. [R. Livengood, et al, J. Vac. Sci. Technol. B 27, 3244 (2009), "Subsurface damage from helium ions as a function of dose, beam energy, and dose rate"] it is reported that 220nm thick Si-membrane can be bent to the extreme curvatures by He-FIB irradiation, Fig. 7(b) in the reference. The role of generated by the irradiation defects in the superplasticity effect should be addressed.

All the references are cited in the new manuscript. The observation of Livengood et al is acknowledged (Paragraph 4 on Page 7). However, elastic properties are always material dependent and the superplasticity of AAO films is reported for the first time in this article.

The role of the irradiation-induced defects is now discussed on page 8, paragraph 1 and 2.

Authors state that the implemented approach allows precise control over the atomic structure, nanoscale, and microscale morphology of the materials. This reviewer sees a trend in structural change, but "precise control" is a strong overestimate.

The relevant statements are now softened. For example "nanometer precision" is used in some cases.

To summarize, this referee can neither conclude, nor exclude the proposed explanations and main conclusions of the paper from the presented data. It may be helpful to make an experiment on freestanding AAO membranes of different thicknesses and under control of low C-deposition. This referee does not recommend the manuscript in present form for publication in the Nature Communications.

The requested experiment were performed under low C-deposition conditions and on free-standing AAO membranes.

Reviewers' comments:

Reviewer #1 (Remarks to the Author):

Thank you for addressing my initial concerns.

I would request one more change or at least consider it.

The term "superplasticity" typically describes a specific phenomena where extremely large strains can be achieved on a large number of materials traditionally related to creep at fine grained materials at elevated temperatures.

Maybe instead of saying simply "superplasticity" it would be better to say 'superplastic like behavior" or "radiation induced superplastic like behavior" to avoid any confusion with the traditional grain boundary sliding at elevated temperatures.

Reviewer #2 (Remarks to the Author):

This referee recommends the paper for publication after a minor revision:

- As a reader I would appreciate a clear definition of "low-flux" and "high-flux" terms already in the beginning of the paper. Currently, the definitions are given on page 4 where Fig.4 is described. The terms appear earlier in Fig.1 and schematic 1.

- Is there a physical meaning of $F = 50 \text{ ions nm}^{-2} \text{ s}^{-1}$? Authors call it the intermediate regime between $F < 10$ and $F > 100$, low and high flux regimes. Is there a connection to a characteristic physical lengthscale in the system or material property that defines this particular number?

- The authors define the Flux in $\text{ions nm}^{-2} \text{ s}^{-1}$. It should be clearly stated in the main text of the paper the exact experimental parameters. The real flux of ions is much higher! and it is scanned over the sample. Typical beam current is 1pA and is focused in to a typical area below 1 nm^2 resulting in the ion flux of $\sim 10^7 \text{ ions nm}^{-2} \text{ s}^{-1}$. Then, this flux with certain speed is scanned over the sample. Authors should state step-size, dwell-time and number of scanned loops (frames). Will result of the experiment be the same if number of scanned frames would be only one (with very long dwell-time)? If authors did not try it, then "Flux" term in the paper has to be redefined. Otherwise, it should be stated that the outcome of experiment is independent of dwell-time and number of scanned frames.

- Fig.3 (e) shows that there is no much Si-EELS signal in the Si substrate. What is the reason? This is crucial since authors state no Si-trace in the AAO layer. If there is no Si-signal from Si substrate, then absence of Si-signal from AAO layer does not prove the absence of Si in the irradiated AAO layer.

- Fig.4 (e) shows a measurement at ambient temperature and at 333K. Could authors describe how this temperature was measured (already partially mentioned in the Supporting info). What is precision of this temperature measurement? Is there an explanation why a small ($\sim 10\%$) change in temperature results in a factor of 2 change in pore closure rate? The change is dramatic.

- Did authors estimate the local temperature of the material at the area of interaction with the ion-beam? For example, It was reported that irradiation by electrons can substantially heat quartz nanopores and actually shrink them by local heating/melting mechanism:

[dx.doi.org/10.1021/nl400304y](https://doi.org/10.1021/nl400304y) , NanoLett. 2013, 13, 1717-1723

“Controllable Shrinking and Shaping of Glass Nanocapillaries under Electron Irradiation”

- As mentioned in the previous report, the effect of superplasticity is not unexpected and has already been discovered in a number of other material systems. Silicon has been already mentioned. Below, is a couple of literature reports on enhanced ductility by He-ions irradiation (this list is not complete):

[dx.doi.org/10.1021/nl502074d](https://doi.org/10.1021/nl502074d) , Nano Lett. 2014, 14, 5176–5183

“Effects of Helium Implantation on the Tensile Properties and Microstructure of Ni₇₃P₂₇ Metallic Glass Nanostructures”

DOI: [10.1021/acs.nanolett.6b00864](https://doi.org/10.1021/acs.nanolett.6b00864) , NanoLett. 2016, 16, 4118-4124

“Radiation-Induced Helium Nanobubbles Enhance Ductility in Submicron-Sized Single-Crystalline Copper”

- So called, channeling effect is well known when charged beams interact with crystalline materials. Could channeling be responsible for low damage of crystalline-AAO film?

Reviewer #1 (Remarks to the Author):

- Thank you for addressing my initial concerns.

I would request one more change or at least consider it.

The term "superplasticity" typically describes a specific phenomena where extremely large strains can be achieved on a large number of materials traditionally related to creep at fine grained materials at elevated temperatures.

Maybe instead of saying simply "superplasticity" it would be better to say 'superplastic like behavior' or "radiation induced superplastic like behavior" to avoid any confusion with the traditional grain boundary sliding at elevated temperatures.

Thanks again for reviewing our manuscript and the positive comments and constructive suggestions.

First of all, we certainly agree that the superplastic-like behavior is indeed induced by radiation, specifically, by ion beams. This is why the title of our manuscript prominently contains the term "ion-irradiation".

To precisely follow the Reviewer's suggestion to articulate **the behavior** type of phenomenon observed, we have changed the title of the relevant section (previously called "Superplasticity") to "Superplastic-like behavior" on page 8.

To make this point even more clear, a note is added on page 8, paragraph 4 to indicate the difference between irradiation-induced plasticity and traditional use of the term:

"It should be noted that the observed superplastic-like behavior is induced by ion-irradiation at *room-temperature* and might be of a different nature compared to the superplasticity commonly attributed to grain boundary sliding in matter subjected to *high temperatures*."

Reviewer #2 (Remarks to the Author):

- This referee recommends the paper for publication after a minor revision:

We thank the Referee for the constructive comments and recommendation for publication (after a minor revision).

We have implemented all the changes requested by the Referee and have made our best effort to satisfactorily and constructively respond to all the questions raised.

- As a reader I would appreciate a clear definition of “low-flux” and “high-flux” terms already in the beginning of the paper. Currently, the definitions are given on page 4 where Fig.4 is described. The terms appear earlier in Fig.1 and schematic 1.

Thanks for this suggestion. Now the definitions appear earlier in the text (page 3) and also in the caption of Figure 1.

Also “low-flux” and “high-flux” are used within quotation marks when appropriate to indicate the relative meaning of the words, which are used to more simply reflect the importance of the flux in various complex observations (implemented in schematic 1, Figure 1 and also the in main text, for example on page 3).

- Is there a physical meaning of $F = 50 \text{ ions nm}^{-2} \text{ s}^{-1}$? Authors call it the intermediate regime between $F < 10$ and $F > 100$, low and high flux regimes. Is there a connection to a characteristic physical length scale in the system or material property that defines this particular number?

We appreciate this question and admit that there is presently no conclusive quantitative answer to it.

At this stage we can only confidently define the low ($F < 10$) and high ($F > 100$) flux regimes as the identifiable thresholds within the limitations of our experiments that show the observed remarkable differences in material response.

The difference between the material responses could be attributed to the ability of the certain amount of ions to carry a certain amount of energy across the nanoporous material to induce the possible rearrangements of the atomic bonds. While certainly this is speculative at this stage, we only emphasize that more fundamental studies are required to determine the irradiation-induced diffusion length scales then one should be able to compare those interaction length scale to the physical length scale of the material mentioned above.

We hope that in our future works we would be able to answer to this important fundamental question.

This point is now highlighted in the last sentence of Conclusion, on page 10:

“More fundamental studies are required to determine the irradiation-induced diffusion length-scales and compare them with the physical length-scales of nanomaterials.”

Also a new relevant reference is added to the manuscript which deals with the spatial correlations and strain responses of materials:

[DOI: 10.1126/science.aai8830, Structure-property relationships from universal signatures of plasticity in disordered solids, Science 2017, 358]

- The authors define the Flux in ions $\text{nm}^{-2} \text{s}^{-1}$. It should be clearly stated in the main text of the paper the exact experimental parameters. The real flux of ions is much higher! and it is scanned over the sample. Typical beam current is 1pA and is focused in to a typical area below 1 nm^2 resulting in the ion flux of $\sim 10^7 \text{ ions nm}^{-2} \text{ s}^{-1}$. Then, this flux with certain speed is scanned over the sample. Authors should state step-size, dwell-time and number of scanned loops (frames). Will result of the experiment be the same if number of scanned frames would be only one (with very long dwell-time)? If authors did not try it, then “Flux” term in the paper has to be redefined. Otherwise, it should be stated that the outcome of experiment is independent of dwell-time and number of scanned frames.

We greatly appreciate this comment and we understand the source of confusion in our definition of “flux” in the manuscript. The referee’s example for ion-beam flux calculation is referring to the flux of the ions arriving the sample at the focus point (let’s call it “pixel flux”). The reported values in our manuscript are corresponding to the “frame flux” which is the “pixel flux” divided to the total number of the pixels in a given scanned frame. (i.e. $F_{\text{pixel}} = \frac{I_0}{A_{\text{pixel}}}$ and

$F_{\text{frame}} = \frac{I_0}{N_{\text{pixel}} \times A_{\text{pixel}}}$, where I_0 is the ion current, A_{pixel} is the pixel area, N_{pixel} is the number of the pixels in a given frame - typically 512×512 pixels in one frame). Therefore, one can always calculate F_{pixel} from F_{frame} , or vice versa.

This point is now reflected in the manuscript, by giving a more precise description to the term “flux” on the last paragraph of page 3.

We now clearly state in the manuscript the exact experimental parameters, including ion current, dwell time, step-size and frames on page 4 paragraph 2.

The experiments that were performed at different dwell times, suggest that the flux-dependent regimes are independent of dwell time. The phenomenological outcome of the both regimes (i.e. lateral mass flow and superplastic regimes) holds regardless of any change in the dwell times (or number of frames). However, increased dwell times may results in anisotropic mass distribution at the exposed areas (i.e. significant mass flow in the prolonged exposed areas), and can negatively influence the symmetry of the final structure. Indeed we used very short dwell times ($0.1 \mu\text{sec}$) to achieve structures with more isotropic character. The nearly symmetrical pore closure is also advantageous in image analysis for pore size estimations. This sentence is added to the 2nd paragraph of page 4:

“These flux-dependent results are phenomenologically independent of scanning parameters (such as scan rate, size or number), however, fastest scanning rate (i.e. shortest dwell-time) was implemented deliberately to achieve more homogeneous and isotropic mass-flow patterns.”

- Fig.3 (e) shows that there is no much Si-EELS signal in the Si substrate. What is the reason? This is crucial since authors state no Si-trace in the AAO layer. If there is no Si-signal from Si substrate, then absence of Si-signal from AAO layer does not prove the absence of Si in the irradiated AAO layer.

Thanks for the comment. We have improved the contrast of the EELS maps for Si (Fig. 3(e)) to clearly show that Si is present. Importantly, Si deposition is not the main reason for pore closure because the pore closure was observed in free-standing membranes without Si.

- Fig.4 (e) shows a measurement at ambient temperature and at 333K. Could authors describe how this temperature was measured (already partially mentioned in the Supporting info). What is precision of this temperature measurement? Is there an explanation why a small (~10%) change in temperature results in a factor of 2 change in pore closure rate? The change is dramatic.

We appreciate this suggestion. In brief, the details of the temperature measurements (with the precision of about 10 °C) are now added, and we also explained that even such small difference in temperature could indeed lead to such significant changes because of the exponential dependence of diffusion rates on temperature. The detailed explanation is below:

The samples were heated in the preloading chamber just before putting the sample into the main chamber for immediate analysis. The temperature in the preloading chamber was measured with an IR thermometer. The sample temperature inside the main chamber is expected to be within 10 °C of the quoted temperature. This point is now included in the relevant section in experimental methods (page 3, paragraph 1 of Supporting Information).

The influence of temperature on irradiation-induced diffusion is expected to be significant [ref: DOI:10.1063/1.3569705, Thermal activation and saturation of ion beam sculpting. J Appl Phys 109, 74312-743124]. Our experiments also confirm the significance of temperature on the irradiation-induced diffusion.

To show that a dramatic temperature-dependence is expected we simply calculate the ratio of the diffusion constants at the given temperatures (333 and 295 K), using a simple formula: [ref: DOI:10.1063/1.3569705, Thermal activation and saturation of ion beam sculpting. J Appl Phys 109, 74312-743124].

$$D(T) = D_0 \exp\left(\frac{-E_a}{k_B T}\right)$$

where $D(T)$ is the temperature dependent diffusion constant, D_0 is a prefactor, E_a is activation energy for surface diffusion (e.g. 0.66 and/or 2.5 eV in a case of Al_2O_3 [ref: Transport phenomena in aluminum oxide, MO Davies - 1965 - ntrs.nasa.gov]), k_B is the Boltzmann constant and T is the temperature. By putting the values in this formula one can obtain $D(333)/D(295) = 7$ and 1000 for $E_a = 0.66$ and 2.5 eV, respectively. Since diffusion length is inversely proportional to diffusion constant, it is expected that the diffusive entities have significantly lower diffusion length (mean-free-path) at higher temperatures which in fact negatively influences the mass-flow rates.

Our experiments shows only a factor of 2 decrease in the mass-flow rate at the higher temperature. One should consider that the calculated values are estimations and also note that the exact correlation of diffusion length and diffusion constant is unknown in this complicated system and that might be the source of errors in comparison between the experiment and calculations.

The effect of reduced “mean-free-path” due to increased temperature is now highlighted in the manuscript on page 6, paragraph 1.

The relevant references are provided in the main text.

- Did authors estimate the local temperature of the material at the area of interaction with the ion-beam? For example, It was reported that irradiation by electrons can substantially heat quartz nanopores and actually shrink them by local heating/melting mechanism:
[dx.doi.org/10.1021/nl400304y](https://doi.org/10.1021/nl400304y) , NanoLett. 2013, 13, 1717-1723
“Controllable Shrinking and Shaping of Glass Nanocapillaries under Electron Irradiation”

We appreciate this question and would like to emphasize that owing to strongly non-equilibrium character of the problem, careful determination of temperature at the interaction spot is not trivial (both technically and conceptually) and is a complex problem going well beyond the scope of this study.

Here, we only note that the melting temperature of Aluminum Oxide is $\sim 2,072$ °C, which seems to be very unlikely to be reached by “cold-ion beam irradiation” (i.e. light ion-beams with keV energies). Moreover, if we assume that temperature rise of the sample is responsible for mass-flow, then higher local temperatures and higher rates of mass-flow are anticipated at higher ion-beam fluxes, which is not the case in our experiment and mass-flow reduces by flux.

Additionally, we have seen that the pore closure rate significantly reduced at temperatures higher than room-temperature (~ 60 °C - well below the melting temperature of alumina), which is inconsistent with the hypothesis of thermally-induced mass-flow.

These points are now highlighted in our manuscript on paragraph 1 of page 6. Readers are now referred to a more detailed discussions on thermal effects induced by electron and ion irradiation in other relevant works (in addition to the article suggested by the referee):

- Johannes, A. *et al.* Anomalous plastic deformation and sputtering of ion irradiated silicon nanowires. *Nano letters* **15**, 3800-3807 (2015).
- Hoogerheide, D. P., George, H. B., Golovchenko, J. A. & Aziz, M. J. Thermal activation and saturation of ion beam sculpting. *J Appl Phys* **109**, 74312-743124, doi:10.1063/1.3569705 (2011).

- As mentioned in the previous report, the effect of superplasticity is not unexpected and has already been discovered in a number of other material systems. Silicon has been already mentioned. Below, is a couple of literature reports on enhanced ductility by He-ions irradiation (this list is not complete):
[dx.doi.org/10.1021/nl502074d](https://doi.org/10.1021/nl502074d) , Nano Lett. 2014, 14, 5176–5183
“Effects of Helium Implantation on the Tensile Properties and Microstructure of Ni73P27 Metallic Glass Nanostructures”
[DOI:10.1021/acs.nanolett.6b00864](https://doi.org/10.1021/acs.nanolett.6b00864) , NanoLett. 2016, 16, 4118-4124
“Radiation-Induced Helium Nanobubbles Enhance Ductility in Submicron-Sized Single-Crystalline Copper”

We are very thankful to the Referee for pointing out on the available relevant knowledge, including the suggested references which we have included in the revised manuscript.

These references helped us to emphasize that superplasticity has indeed been reported for a number of other materials due to irradiation-induced bubble formation.

A note is added in the manuscript (see page 8):

“Also the observed phenomena are of a different nature compared to the enhanced ductility of materials due to irradiation-induced bubble formation.”

- So called, channeling effect is well known when charged beams interact with crystalline materials. Could channeling be responsible for low damage of crystalline-AAO film?

Thanks for the comment. Indeed we considered channeling effect in irradiated c-AAO by varying the relative angle of the impinging ions. The resistant character of c-AAO against irradiation was observed at different irradiation angles. Since channeling effect is highly sensitive to the crystal orientation of the target material (i.e. the relative angle between the crystal orientation and the incident beam), it is reasonable to assume that channeling is not the major influential parameter in irradiation resistivity of c-AAO. This point is now reflected in the manuscript:

Page 7, paragraph 1: *“Ion-channelling in crystalline materials is another source of resistivity against damage in irradiation experiments, however, since irradiation resistivity of c-AAO is independent of the incident angle of the ion-beam, it is reasonable to assume that ion-channelling is not the major influential parameter in irradiation resistivity of c-AAO.”*

REVIEWERS' COMMENTS:

Reviewer #2 (Remarks to the Author):

This referee would like to thank authors for the clarifications and careful answers to the questions. The manuscript is recommended for publication in the current form.